# *Lentinula edodes* Cultured Extract and *Rouxiella badensis* subsp. *acadiensis* (Canan SV-53) Intake Alleviates Immune Deregulation and Inflammation by Modulating Signaling Pathways and Epigenetic Mechanisms

**DOI:** 10.3390/ijms241914610

**Published:** 2023-09-27

**Authors:** Roghayeh Shahbazi, Hamed Yasavoli-Sharahi, Nawal Alsadi, Farzaneh Sharifzad, Sandra Fang, Cyrille Cuenin, Vincent Cahais, Felicia Fei-Lei Chung, Zdenko Herceg, Chantal Matar

**Affiliations:** 1Cellular and Molecular Medicine Department, Faculty of Medicine, University of Ottawa, Ottawa, ON K1H 8M5, Canada; rshah017@uottawa.ca (R.S.); hyasa068@uottawa.ca (H.Y.-S.); nalsa068@uottawa.ca (N.A.); 2Department of Urology, Feinberg School of Medicine, Northwestern University, Chicago, IL 60611, USA; farzaneh.sharifzad@northwestern.edu; 3Translational Molecular Medicine Department, Faculty of Medicine, University of Ottawa, Ottawa, ON K1H 8M5, Canada; sfang075@uottawa.ca; 4Epigenomics and Mechanisms Branch, International Agency for Research on Cancer (IARC), 25 Av. Tony Garnier, 69007 Lyon, France; cueninc@iarc.fr (C.C.); cahaisv@iarc.fr (V.C.); feliciacfl@sunway.edu.my (F.F.-L.C.); hercegz@iarc.fr (Z.H.); 5Department of Medical Sciences, School of Medical and Life Sciences, Sunway University, Jalan Universiti, Bandar Sunway, Subang Jaya 47500, Selangor, Malaysia; 6School of Nutrition, Faculty of Health Sciences, University of Ottawa, Ottawa, ON K1H 8M5, Canada

**Keywords:** gut microbiota, *Lentinula edodes* mycelia, probiotic *Rouxiella badensis*, interleukin-17, FOXO1, STAT3, microRNA, DNA methylation

## Abstract

Puberty is a critical developmental period of life characterized by marked physiological changes, including changes in the immune system and gut microbiota development. Exposure to inflammation induced by immune stressors during puberty has been found to stimulate central inflammation and lead to immune disturbance at distant sites from the gut; however, its enduring effects on gut immunity are not well explored. Therefore, in this study, we used a pubertal lipopolysaccharides (LPS)-induced inflammation mouse model to mimic pubertal exposure to inflammation and dysbiosis. We hypothesized that pubertal LPS-induced inflammation may cause long-term dysfunction in gut immunity by enduring dysregulation of inflammatory signaling and epigenetic changes, while prebiotic/probiotic intake may mitigate the gut immune system deregulation later in life. To this end, four-week-old female Balb/c mice were fed prebiotics/probiotics and exposed to LPS in the pubertal window. To better decipher the acute and enduring immunoprotective effects of biotic intake, we addressed the effect of treatment on interleukin (IL)-17 signaling related-cytokines and pathways. In addition, the effect of treatment on gut microbiota and epigenetic alterations, including changes in microRNA (miRNA) expression and DNA methylation, were studied. Our results revealed a significant dysregulation in selected cytokines, proteins, and miRNAs involved in key signaling pathways related to IL-17 production and function, including IL-17A and F, IL-6, IL-1β, transforming growth factor-β (TGF-β), signal transducer and activator of transcription-3 (STAT3), p-STAT3, forkhead box O1 (FOXO1), and miR-145 in the small intestine of adult mice challenged with LPS during puberty. In contrast, dietary interventions mitigated the lasting adverse effects of LPS on gut immune function, partly through epigenetic mechanisms. A DNA methylation analysis demonstrated that enduring changes in gut immunity in adult mice might be linked to differentially methylated genes, including *Lpb*, *Rorc*, *Runx1*, *Il17ra*, *Rac1*, *Ccl5*, and *Il10*, involved in Th17 cell differentiation and IL-17 production and signaling. In addition, prebiotic administration prevented LPS-induced changes in the gut microbiota in pubertal mice. Together, these results indicate that following a healthy diet rich in prebiotics and probiotics is an optimal strategy for programming immune system function in the critical developmental windows of life and controlling inflammation later in life.

## 1. Introduction

Dynamic interactions between commensal bacteria living in the intestine and the host are crucial for the development and function of the immune system and intestinal homeostasis [1]. Any alteration to the composition of the commensal population in the gut that negatively affects mutualistic relationships among microbial communities is considered dysbiosis. Uncorrected dysbiosis may lead to various chronic inflammatory disorders, including inflammatory bowel disease (IBD), obesity, and diabetes [2,3].

Exposure to immune stressors such as exogenous lipopolysaccharide (LPS) is accompanied by gut dysbiosis and permeability [4,5,6] and mediates production of inflammatory cytokines such as interleukin (IL)-6 and IL-23 by immune cells [7]. These cytokines, along with IL-1β, enhance IL-17 production in the presence of T cell receptor (TCR) stimulation by inducting T helper (Th)-17 cells differentiation [8,9,10,11,12]. Th17 cells are the major source of IL-17 production in adaptive immunity [9]. In addition, innate immune cells such as γδ T cells and type 3 innate lymphoid cells (ILC3s) secrete IL-17, in response to IL-1 and IL-23, without TCR engagement [9,13]. IL-6 and IL-23 favor IL-17 production through signal transducer and activator of transcription 3 (STAT3) signaling [14,15]. IL-1β initiates the IL-6/STAT3 pathway [16]. Evidence shows that IL-6/STAT3 inhibits forkhead box O (FOXO)-1 expression [17]. FOXO1 suppresses Th17 differentiation and pathogenicity, and IL-17A and IL-17F expression [17]. IL-17 is known for its ability to initiate inflammation and autoimmunity by induction of a variety of cytokines such as IL-6 and IL-1β [18,19].

Additionally, gut commensal bacteria regulate IL-17 production [20]. Gut microbiota-derived short-chain fatty acids control IL-17A production by repressing IL-17-producing γδ T cells [20]. Segmented filamentous bacterium is a gut commensal that was found to favor the proliferation of Th17 cells [12] and stimulate the expression of IL-17A in CD4^+^ T cells in the lamina propria [21]. In addition, epigenetic modifications such as DNA methylation and histone modification may regulate IL-17-producing cell differentiation, including Th17 and γδ T cells differentiation [22,23]. Different cytokines such as IL-6, IL-1β, IL-23, and TGF-β may contribute to the epigenetic regulation of IL-17 expression [24]. Also, gut microRNAs (miRNAs) regulate the intestinal immune system [25]. Dysregulation of the gut miRNA profile correlates with inflammatory responses [26]. It has been shown that gut miRNAs regulate IL-17 production by modulating intestinal Th17 cell response [27].

A growing body of evidence has shown the health-promoting benefits of prebiotics and probiotics, related to their anti-inflammatory and immunomodulatory activities at the gut level and beyond [28,29]. Manipulating the gut microbiota by probiotics could be used as a strategy for maintaining gut microbial composition, gut barrier function, and gut immune system homeostasis and preventing diseases related to chronic inflammation [30,31]. Prebiotics favor probiotic commensal bacteria growth and eliminate pathogens. Prebiotics also directly act on the gut epithelial cells and innate immunity by regulating the Toll-like receptors (TLRs) and inflammatory signaling pathways [31].

Puberty is a critical developmental window of life characterized by marked physiological changes [32]. Gut microbiota development continues throughout adolescence [33]. The immune system as a potential mediator of developmental programming undergoes profound alteration throughout puberty [34]. It has been shown that pubertal LPS exposure can induce dysbiosis, program the peripheral and central immune responses, and affect brain function [35,36]; however, its lasting effects on gut immunity are not well explored. Probiotic intake might mitigate pubertal LPS-mediated behavioral changes [36], but the effect of prebiotic and probiotic intake on mitigating enduring LPS-induced inflammatory response at the gut level is not well studied.

Therefore, the current study aimed to investigate whether exposure to LPS in the pubertal window results in immune dysfunction in mice during early adulthood and whether dietary intervention during puberty can block LPS-induced immune deregulation later in life by exploring selected cytokines, signaling pathways, and epigenetic mechanisms related to IL-17 production and function. To this end, pubertal female Balb/c mice were exposed to LPS and fed a prebiotic compound, a standardized extract of cultured *Lentinula edodes* mycelia (ECLM) referred to as AHCC or probiotic bacterium *Rouxiella badensis* subsp. *acadiensis* (Canan SV-53) referred to as SV-53, to study the effects of treatment on the immune system at the gut level.

## 2. Results

### 2.1. Effect of the Treatment on the Cytokine Concentrations in the Small Intestine

First, we examined the immediate inflammatory effects of LPS exposure and prebiotic/probiotic intake on immune system function by measuring IL-17A and IL-17F levels in the small intestine of pubertal mice. The IL-17 cytokine family consists of six members, including IL-17A, IL-17B, IL-17C, IL-17D, IL-17E, and IL-17F. Among all the members, IL-17A and IL-17F share the highest sequence homology and are best studied [9]. Dysregulated IL-17A and IL-17F contribute to chronic inflammation and autoimmunity [9]. The concentration of IL-17A and IL-17F was higher in mice challenged with a single dose of LPS in puberty than in mice receiving AHCC and AHCC + LPS (*p* < 0.05) (Figure 1A). Then, we measured the levels of a variety of cytokines related to IL-17A production, such as TGF-β, IL-6, IL-1β, IL-23, and IL-10 (Figure 1A). The TGF-β level was higher in the AHCC group compared to the control (*p* < 0.01), LPS (*p* < 0.001), and AHCC + LPS (*p* < 0.01). LPS significantly induced IL-6 secretion (*p* < 0.05) and AHCC intake decreased the effect of LPS on IL-6 concentration but did not reach significance. The IL-1β concentration was significantly lower in the AHCC group compared to the other groups (*p* < 0.05 vs. control and AHCC + LPS and *p* < 0.01 vs. LPS). The IL-23 level was also lower in the AHCC group compared to other groups (*p* < 0.05 vs. control and AHCC + LPS and *p* < 0.001 vs. LPS). The level of IL-10 was elevated in groups challenged with LPS compared to unchallenged mice receiving AHCC (*p* < 0.01).

Next, the enduring effect of pubertal exposure to LPS and prebiotic intake on the same cytokine profile was examined in adult mice (Figure 1B). LPS exposure during puberty caused an enduring increase in IL-17A levels in comparison with control mice (*p* < 0.01), while prebiotic intake mitigated the impact of LPS on IL-17A (*p* < 0.05). In AHCC + LPS mice, the IL-17F level was lower than in the control and LPS groups (*p* < 0.05). The TGF-β level was higher in AHCC + LPS mice compared to control and LPS mice (*p* < 0.05). The AHCC intake in mice challenged with LPS was correlated with a lasting reduction in IL-6 concentration, compared to mice that received pubertal LPS without receiving AHCC (*p* < 0.05). Feeding mice with AHCC mitigated the stimulatory impact of LPS on IL-1β (*p* < 0.05). No significant difference was seen in IL-23 concentration among groups. The IL-10 level was lower in LPS mice compared to control and prebiotic groups (*p* < 0.05).

Figure 1C illustrates the probiotic SV-53’s impact on the enduring consequences of LPS on immune responses. Probiotic administration to the LPS-treated mice mitigated the enduring effect of LPS on IL-17A (*p* < 0.05). The IL-17F concentration was significantly lower in the probiotic+LPS group compared to the control and LPS groups (*p* < 0.05). A significant increase in the TGF-β levels was seen in mice receiving probiotics compared to untreated mice (*p* < 0.01) and mice exposed to LPS (*p* < 0.0001). Although probiotic intake decreased the inhibitory effect of LPS on TGF-β, the level of this cytokine remained lower in the probiotic+LPS group than in the probiotic group (*p* < 0.01). Probiotic intake also inhibited the stimulatory effect of LPS on IL-6 (*p* < 0.05). No significant difference was observed in IL-23 levels among groups. The IL-10 concentration was lower in adult mice challenged with pubertal LPS compared to unchallenged and SV-53 mice (*p* < 0.05).

### 2.2. Effect of the Treatment on p-STAT3, STAT3, and FOXO1 Levels in the Small Intestine

To understand the underlying mechanisms governing the immunoprotective effects of the treatments, both the acute and enduring effects of treatments on p-STAT3, STAT3, and FOXO1 levels were studied. IL-17 production by the lamina propria CD4^+^ T cells is dependent on STAT3 activation [37], while FOXO1 is a negative regulator of Th17 differentiation and IL-17 production [38]. In the puberty window, LPS challenge increased the level of p-STAT3 as compared to the control and AHCC groups (*p* < 0.05). AHCC intake could inhibit the stimulatory effect of LPS on p-STAT3 (Figure 2A). LPS challenge had no significant acute effect on STAT3 (Figure 2A). Also, in pubertal mice, a significant increase in the FOXO1 level was found in mice receiving AHCC for one week compared to the control (*p* < 0.01) (Figure 2A). In adult mice, p-STAT3 levels were significantly lower in AHCC and AHCC + LPS mice than in LPS mice (*p* < 0.01 and *p* < 0.05, respectively) (Figure 2B). In addition, a lasting increase in STAT3 production was observed in LPS mice compared to the control mice (*p* < 0.05), while AHCC intake for two weeks significantly inhibited the effect of LPS on STAT3 (*p* < 0.01) (Figure 2B). Regardless of LPS exposure, prebiotic consumption during puberty stimulated FOXO1 production in adult mice (*p* < 0.05) (Figure 2B).

Furthermore, in mice challenged with LPS at puberty, probiotic SV-53 administration was able to prevent the stimulatory effect of LPS on p-STAT3 and STAT3 in adult mice (*p* < 0.05 and *p* < 0.01, respectively) (Figure 2C). An increase in the FOXO1 protein was found in the probiotic group compared to the control (*p* < 0.05) and LPS (*p* < 0.01) groups, and in the SV-53+LPS mice compared to the LPS counterparts (*p* < 0.05) (Figure 2C).

### 2.3. Effect of the Treatment on the Gut Microbiota of Mice in Pubertal Window

To study the effect of treatment on microbial richness and evenness, different metrics, including observed features, Chao1, Shannon, and Simpson indices, were calculated to assess alpha diversity. At the level of 97% similarity, varied alpha metric results showed no significant difference in alpha diversity. Figure 3A illustrates the alpha diversity inferred by the Shannon index plot. For beta diversity, a principal coordinates analysis (PCoA) based on the weighted UniFrac Matrix was used to visualize group differences (Figure 3B). Then, the beta diversity quantitative distance metric was performed to assess the significance of differences between the groups using the PERMANOVA pairwise test. Our results indicated a significant difference between LPS and prebiotic (Pseudo-F: 5.859, adjusted *p*-value or *q*-value: 0.03) and between LPS and prebiotic+LPS (Pseudo-F: 3.959, *q*-value: 0.03) groups (Figure 3C).

Tannerellaceae, Bacteroidaceae, and Flavobacteriaceae were the most abundant families seen in all groups (Figure 3D). A differential abundant analysis revealed no significant difference between control and treated groups at the various taxonomic levels. We found differential abundant taxa between LPS and prebiotic, and between LPS and prebiotic+LPS groups at the various levels (Figure 3E). Most importantly, the abundance of Bacteroides and Parabacteroides genera, and the abundance of *Bacteroides intestinalis* (*B. intestinalis*) species was markedly higher in mice that received only LPS compared to mice that received only AHCC. The same results at the genus level were obtained when comparing the LPS and AHCC + LPS groups, indicating the potential of prebiotic intake in alleviating LPS’s impact on gut bacterial communities (Figure 3E). AHCC intake one week before the LPS challenge decreased the impact of LPS on *B. intestinalis*, without reaching a significant level (Figure 3E).

### 2.4. Effect of the Treatment on the miR-145 and miR-425 Expressions in the Small Intestine

miR-145 has been shown to improve chronic inflammatory diseases by targeting various proteins [39]. FOXO1 enhances the miR-145 expressions [40,41]. In addition, miR-145 may suppress STAT3 activation in cancer [42]. Inhibition of FOXO1 by miR-425 has been reported [43]. miR-425 also increases pathogenic Th17 differentiation and IL-17 production [43]. We have previously shown the role of a polyphenolic mixture fermented by SV-53 in the chemoprevention of mammary carcinoma and controlling breast cancer stem cells by increasing miR-145 and FOXO1 [41,44]. Therefore, in the current research, we studied the effect of the treatment on miR-145 and miR-425 expression. LPS and prebiotic intake did not affect miR-145 expression in mice at puberty (Figure 4A); however, a significant decrease in miR-425 expression was observed in the AHCC + LPS group compared to LPS (*p* < 0.01) (Figure 4A). In adult mice, receiving AHCC during puberty correlated with an enduring elevation in miR-145 expression when compared to control (*p* < 0.0001 for AHCC, and *p* < 0.001 for AHCC + LPS groups) and LPS (*p* < 0.001 for AHCC, and *p* < 0.05 for AHCC + LPS groups) (Figure 4B). In mice challenged with pubertal LPS, AHCC intake led to an enduring reduction in miR-425 compared to the LPS group (*p* < 0.01) (Figure 4B). Moreover, feeding mice with probiotic SV-53 upregulated miR-145 expression compared to control (*p* < 0.05) and LPS-injected mice (*p* < 0.0001) (Figure 4C). In mice challenged with LPS, probiotic intake reduced the inhibitory effect of LPS on miR-145 expression (*p* < 0.05) (Figure 4C). We found a significant downregulation in miR-425 expression in the probiotic LPS group compared to the control and LPS groups (*p* < 0.05) (Figure 4C).

### 2.5. Effect of Treatment on DNA Methylation Status in Small Intestine of Mice

The methylation status of the regions around the transcriptional start sites (TSS) is important for gene expression. DNA hypermethylation within TSS is associated with gene expression repression [45,46].

To determine if the lasting effects of pubertal challenge with LPS and prebiotic intake on immune system function are mediated partly through epigenetic modifications, the methylation status of genes in the ileum samples of adult mice in AHCC + LPS vs. LPS (Figure 5) and LPS vs. control (Appendix A) were examined. By applying the cut-off of absolute group mean difference in beta values of >0.05 and FDR adjusted *p*-value < 0.05, differentially methylated probes (DMPs) were identified and were used as a basis for regional analysis where we found statistically significant differentially methylated regions (DMRs) related to various immune and metabolism pathways such as MAPK, PI3K-Akt, JAK-STAT, and T cell receptor signaling pathways. Figure 5A and Appendix A represent the distribution of DMRs within different genomic regions, and Figure 5B and Appendix A illustrate the top enriched pathways for genes associated with the identified DMRs. Notably, when focusing on genes related to the objectives of our study, a significant hypermethylation of CpGs was identified within the promoter of the LPS-binding protein gene (*Lbp*) in LPS-challenged and AHCC-fed mice in comparison to LPS-challenged mice (Figure 5C). We also found significant hypermethylation of promoter regions of genes directly controlling IL-17A production or function, including RAR-related orphan receptor C (*Rorc*), runt-related transcription factor 1 (*Runx1*), and IL-17 receptor A gene (*Il17ra*) in AHCC + LPS mice compared to their LPS counterparts (Figure 5C). Here, 1 to 5 kb regions within the chemokine (C-C motif) ligand 5 (*Ccl5*) and Rac family small GTPase 1 (*Rac1*) were also significantly methylated in AHCC + LPS (Figure 5C). In addition, the promoter of *Il10* and 1 to 5 kb region of the nuclear factor of activated T cells c1 (*Nfatc1*) were hypomethylated in the AHCC + LPS group compared to the LPS group (Figure 5C).

In LPS vs. control mice, hypomethylation of *Runx1*, *Il17ra*, *Rac1*, and *Ccl5*, and hypermethylation of *Il10* and *Nfatc1* genes around TSS were observed; however, by applying the cut-off, only *Rac1*, *Ccl5*, and *Il10* were significantly different between the groups (Appendix A). None of these genes were found among the detected differentially methylated genes when the AHCC + LPS group was compared to the control.

## 3. Discussion

Disruption of gut microbiota by an immune stressor (inflammatory LPS) during puberty has been shown not only to transiently perturb gut microbiota but also to induce immune alterations that could significantly affect developmental programming at distant sites from the gut [47]. Even after microbiota recovery, phenotypic changes such as immune and metabolic changes remain with alterations in long-term developmental programming [48,49]. However, dietary interventions can mitigate these changes and prevent immune perturbation [6]. Probiotics and prebiotics are an integral part of a balanced diet, such as the Mediterranean diet which is well-known as an effective dietary pattern against dysbiosis [50].

In the present study, we investigated the potential role of a prebiotic compound, AHCC, and a probiotic bacterium, SV-53, in mitigating lasting inflammatory immune responses in mice exposed to inflammation and dysbiosis during the developmental period of puberty through regulation signaling pathways and epigenetic mechanisms. Both AHCC and SV-53 have been shown to protect against inflammatory challenges by potentially mimicking immunobiotic effects [36,51]. The prebiotic AHCC is a cultured mushroom mycelium extract shown to favorably modulate the immune system and alleviate cancer burden. AHCC plays an immunomodulatory role by priming the TLR-2 and TLR-4 at the intestinal epithelium [51]. Evidence also suggests that AHCC exerts inhibitory activity on cancer stem cells by modulating miRNAs involved in immune evasion [52]. The probiotic SV-53 is a novel Gram-negative probiotic bacterium isolated from the microbiota of wild blueberry fruit which exerts significant immunomodulatory effects at the gut level by improving gut mucosal immunity and gut barrier integrity, regulating IgA, IL-10, and IL-6 secretion, modulating gut miRNA expression, and reducing the population of pathogenic bacteria such as *Salmonella Typhimurium* and *Staphylococcus aureus* in intestinal fluid [53,54,55,56].

In the current paper, we first studied the immediate effect of LPS on cytokine profiles eight hours post-exposure in pubertal mice, and the role of AHCC intake in inhibiting LPS effects. The effect of LPS treatment on increasing inflammatory cytokine levels shortly after exposure in the serum and tissues has been reported [6,57]. A difference in the acute immune responses to LPS exposure has been found between puberty and adulthood, indicating that immune responses to immune stressors are highly affected by age [58]. We found that prebiotic administration diminished the immediate effects of pubertal LPS exposure on IL-17A and F, TGF-β, IL-6, IL-1β, and IL-23.

Along with the acute effect, the long-term impact of pubertal LPS and prebiotic/probiotic exposure on immune responses in adult Balb/c mice was also studied. Interestingly, in adult mice, prebiotic intake during puberty diminished the lasting stimulatory effect of pubertal LPS challenge on IL-17A, IL-1β, and IL-6 production. Also, in adulthood, the level of TGF-β was higher in mice concomitantly exposed to LPS and AHCC in the pubertal window. Although LPS exposure immediately increased the IL-10 production, in adult mice the levels of IL-10 dropped compared to untreated and prebiotic-treated mice. LPS and dietary intervention had no long-term impact on IL-23 levels. We also observed that probiotic SV-53 intake contracted LPS impact on IL-17A, TGF-β, and IL-6 production. LPS exposure exerted an enduring inhibitory impact on IL-10 production, while probiotic intake elevated IL-10 levels in mice exposed to LPS but did not reach significance.

Based on the results from the cytokines profile, the lasting impact of the treatment on STAT3, p-STAT3, and FOXO1 levels was studied in the small intestine of adult mice. Studies have shown the effect of natural products on the inhibition of IL-17 secretion by targeting the STAT3 pathway [59]. In a study, feeding mice with a *Lactobacillus acidophilus* led to the alleviation of colitis-associated inflammatory responses related to the IL-23/Th17 pathway through inhibition of STAT3 and secretion of IL-17 [59]. In the acute phase, AHCC intake could decrease the stimulatory effect of LPS on p-STAT3 in pubertal mice. In the long term, our result showed the significant role of prebiotic/probiotic intake in mitigating an LPS-derived increase in STAT3 and p-STAT3 in the small intestine of adult mice. Moreover, prebiotic/probiotic administration caused a lasting increase in the intestinal level of FOXO1.

The cytokine’s milieu secretion following antigen recognition by innate immune cells activates different transcription factors such as STAT3 and instructs naïve T cells to acquire effector or regulatory phenotypes [60]. The master transcription factor of Th17 cells, retinoic acid receptor-related orphan receptor-gamma-t (ROR*γ*t), regulates Th17 cell differentiation and is necessary to induce IL-17A expression [61]. In the gut, IL-6 promotes STAT3-dependent expression of RORγt and subsequent Th17 cell proliferation and IL-17 expression [62], while IL-1 and IL-23 induce delayed differentiation and expansion of Th17 cells [8,24]. Studies have shown that the deletion of STAT3 in CD4^+^ T-cells protects mice against the development of experimental autoimmune diseases and suppresses the production of IL-17-expressing T cells [63]. In addition, p-STAT3 activates the transcription of the STAT3 gene, resulting in an accumulation of unphosphorylated STAT3. Unphosphorylated STAT3, in turn, triggers a second wave of cytokine production, specifically IL-6, causing prolonged cytokine-dependent signaling at later stages [64]. On the other hand, IL-10 signaling in T cells blocks the emergence of IL-17A-producing cells [65]. IL-10, along with TGF-β, stimulates regulatory T cell (Treg) development and limits Th17 cell-induced inflammation in the gut [66].

Some evidence indicates that IL-6 and IL-23 in the presence of IL-1β efficiently promote IL-17 secretion from naïve CD4^+^ T cells, possibly by histone modification of the *Il17a*/*Il17f* and *Rorc* promoters directed in the absence of TGF-β1, while TGF-β1 suppresses this polarization [24]. IL-1 signaling in T cells not only promotes early Th17 cell differentiation but also is critical for the maintenance of Th17 cells in the lack of TCR stimuli [67]. IL-1β facilitates IL-17 A and F expression by promoting STAT3 phosphorylation and its binding to key cis-elements that control *Il*17a/f transcription [68]. Our results indicate that LPS may affect IL-17 production, in part, through IL-1β signaling, while prebiotic intake may diminish this effect by inhibiting IL-1β. In a study, systemic LPS administration to mice did not increase Th17 differentiation but drove pre-committed IL-17A-producing Th17 cell expansion independent of IL-23 and possibly through an IL-1 and IL-6-dependent manner [69].

On the other hand, overexpression of IL-17A, in turn, drives the production of IL-6, IL-23, and IL-1β resulting in inflammation amplification [8,70]. An IL–17–stimulated release of IL-6 activates the STAT3 pathway and promotes the differentiation and maturation of Th17 cells and further IL-17 production [70].

In the present research, at the acute phase, a rapid elevation in IL-17 levels in LPS-challenged mice may also indicate the role of innate immune cells such as ILC3s in IL-17 induction, which directly happens in response to IL-1β and IL-23 without further differentiation of CD4^+^ T cells [13].

The FOXO1 transcriptional factor is highly expressed in Treg which mediates the expression of IL-10 and TGF-β [71,72] and suppresses Th17 cell differentiation by binding to ROR*γ*t via its DNA binding domain and inhibiting its transcriptional activity, as well as decreasing IL-17A production and IL-23 receptor expression [38,71]. LPS activates Akt signaling, leading to FOXO1 phosphorylation and inactivation [72]. Moreover, in a study, IL-6/STAT3 signaling highly stimulated miR-183 cluster (miR-183C) expression. This miRNA cluster is connected to the pathogenicity of Th17 due to pathogenic cytokine production, including IL-17A and IL-17F, and mediates autoimmunity by repression of FOXO1 and induction of IL-1R1 expression [17]. Of interest, TGF-B inhibited the IL-6/STAT function in inducing miR-183C in Th17 cells and reduced their pathogenicity [17]. Hence, our results may indicate that prebiotic and probiotic intake exerts immunomodulatory activity partly by modulating STAT3 and FOXO1 signaling and regulating the expression of cytokines that control these pathways, including IL-17A, IL-1β, IL-6, and TGF-β.

Furthermore, we studied the effect of AHCC on the gut microbiota of pubertal mice eight hours after LPS injection. We have previously shown that the administration of LPS and SV-53 to pubertal mice changed gut microbiota in a sex-dependent manner [6]. In the current research, the abundance of Bacteroides and Parabacteroides genera was markedly higher in mice that received only LPS compared to mice in AHCC and their AHCC + LPS counterparts. An increase in Bacteroides and Parabacteroides was positively correlated with colitis-induced mucosal injury [73]. Also, a positive correlation between the augmentation of Parabacteroides and Bacteroides genera and cytokines involved in the pathogenesis of immune-related disorders, including IL-17, IL-21, and INF-γ, has been reported in patients [74].

We found a higher abundance of *B. intestinalis* in the gut microbiota of mice challenged with LPS compared to prebiotic-fed mice, and an insignificant decline in *B. intestinalis* in AHCC + LPS compared to LPS groups. A recent study revealed the critical role of *B. intestinalis* and IL-1β signaling in immune checkpoint blockade (ICB) therapy-induced ileitis [75]. Andrews et al. (2021) found that ICB-induced intestinal inflammation is strongly related to overexpression of ileal *Il1b*. A gut microbiota analysis revealed a marked elevation of *B. intestinalis* in patients with ICB toxicity, along with an overexpression of mucosal IL-1β in patient samples of colitis and mice. Furthermore, colonizing mice with *B. intestinalis* upon microbiota ablation by antibiotics was accompanied by an induction of ileal *Il1b* transcription and increased ileal damage [75].

To investigate whether the persisting effects of LPS exposure and prebiotic intake on gut immunity are mediated by durable epigenetic modifications, we studied miRNA expression and DNA methylation status in the small intestine of adult mice. The epigenetic mechanism regulates microbiota–host interactions [76]. miRNAs derived from intestinal epithelial cells modulate the composition of the intestinal microbial communities [25]. Commensal microbes can regulate DNA methylation by providing primary substrates [77].

In this study, it was shown that prebiotic and probiotic intake stimulated miR-145 expression. Also, feeding LPS-exposed mice with the prebiotic/probiotic significantly diminished the relative expression level of miR-425. Similarly, in a study using a murine model of colorectal cancer, the administration of the probiotic *Bifidobacterium longum* was correlated with an increase in miR-145 and a decrease in IL-6 concentration [78]. Down-regulation of miR-145 is associated with an increase in pathogenic Th17 cell responses [79]. In a study, an over-expression of miR-145 improved metabolic inflammation in mice and prevented LPS-induced inflammation in vitro [39]. Also, miR-145 knockdown was associated with an increased secretion of TNF-α, IL-6, and IL-1β in vitro [80]. In bladder cancer cells, miR-145 expression inhibited STAT3 activation, stimulated FOXO1 expression, and suppressed cell growth [42]. Upregulation of miR-425 has been reported in inflammatory conditions, such as some types of cancers, which is correlated with cancer progression [81,82]. An overexpression of miR-425 promotes Th17 cell production and IL-17A secretion by suppressing FOXO1 in colitis mice [43]. IL-1β was found to induce miR-425 expression in gastric cancer cells by activating NF-κB signaling [83].

Additionally, a DNA methylation analysis revealed an increase in CPGs methylation around TSS of *Lbp*, *Rorc*, *Runx1*, *Il17ra*, *Rac1*, and *Ccl5* in adult mice that received AHCC + LPS during puberty, compared to the mice only challenged with LPS, which may indicate the transcriptional repression of these genes by AHCC.

TLR4 recognizes LPS with the help of diverse proteins such as LBP. LBP binds to LPS molecules and delivers them to CD14. CD14 subsequently transfers LPS to the TLR4/MD-2 complex, which leads to inflammatory signal initiations and proinflammatory cytokine production [84].

*Rorc* encodes RORγt [85]. A conditional deletion of *Rorc* in IL-17A-producing Th17 cells has revealed the critical role of RORγt in Th17 cell stability [86]. Moreover, genetic ablation of *Rorc* in mature Th17 has been found to suppress their pathogenic activity [87]. An overexpression of *Runx1* is associated with the pathogenesis of autoimmunity through Th17 cell induction [88]. Runx1 promotes Th17 generation by enhancing RORγt expression and forming a complex and interacting with RORγt, which upregulates *Il17* transcription [89]. IL-17RA is required for IL-17 cytokines signaling. IL-17/IL-17RA signaling regulates different inflammatory pathways, resulting in the release of pro-inflammatory cytokines such as IL-1β and IL-6 [90]. IL-17/IL-17RA signaling exerts a pathogenic role in multiple inflammatory and autoimmune diseases and targeting this signaling is a remarkable strategy in the treatment of autoimmunity [91,92].

Evidence demonstrates that LPS may induce NF-κB and proinflammatory cytokine response through a Rac1-dependent pathway [93]. Rac1 is necessary for IL-17A expression and induction of autoimmunity in mice. A reduction in IL-17A, IL-17F, and IL-22 levels, and IL-23 receptor expression in Rac1-deficient Th17 cells, has been reported [94]. Moreover, LPS induces CCL5 expression in macrophages through a TLR4-dependent pathway [95], and a higher expression of proinflammatory chemokines, including CCL5, by pathogenic Th17 cells has been reported [96].

In addition, the *Il10* promoter was hypomethylated in the AHCC + LPS group, which may indicate transcription activation of the *Il10*. Evidence shows the protective function of IL-10 against autoimmunity by controlling pathogenic Th17 differentiation [97]. In colitis mice, IL-10 potently suppresses the pro-IL-1β production transcriptionally in macrophages and its maturation to IL-1β, and alters Th17 cytokine dependency required for colitis pathogenesis [98]. IL-10 preserves the FOXO1 function by inhibiting the Akt pathway [66].

NFATc1 is a member of the NFAT family of transcription factors which is activated following TCR stimulation and subsequent dephosphorylation and nuclear import. NFATc1 contributes to Th17 cell differentiation and its deficiency is correlated with a reduced expression of RORγt and Th17-related cytokines such as IL-17A, IL-17F, and IL-21 [99]. However, we observed higher methylation of *Nfatc1*, the gene coding NFATc1 protein, in AHCC + LPS mice than in LPS mice.

Our study has some limitations. Although our results, particularly the p-STAT3, FOXO1, and DNA methylation results, may suggest the role of AHCC in regulating immune responses in inflammatory conditions, in part, by modulating Th17 cells, fluorescence-activated cell sorting or an FACS analysis could be instrumental in determining the different immune cells’ involvement in the anti-inflammatory activity of AHCC. Moreover, although sex difference in immune responses in mice has been previously demonstrated [6], we only studied female mice in this paper because this research is part of a larger study on the gut-mammary gland axis utilizing female Balb/c mice. We also did not perform DNA methylation for the probiotic SV-53 experiment.

In conclusion, we found that exposure to an immune stressor in critical windows of development that alters gut microbiota may cause enduring dysregulation in the cytokines and central proteins involved in gut immunity, in part, by epigenetic modification and, therefore, can lead to lasting immune system dysfunction at the gut level; prebiotic or probiotic intake may mitigate these negative consequences. Together, these results may indicate that following a healthy dietary pattern such as a diet rich in prebiotics and probiotics could be a useful strategy in priming the immune system in early life and preventing health problems later in life.

## 4. Materials and Methods

### 4.1. Animals

Four-week-old female Balb/c mice weighing 13–17 g (Charles River, Montreal, QC, Canada) were used in the current study. Three mice were housed together in plastic cages in a controlled atmosphere (temperature 22 ± 2 °C; humidity 55 ± 2%) with a 12 h light/dark cycle. During the study, all groups received a conventional balanced diet ad libitum. Mice were maintained and treated following the guidelines of the Canadian Council on Animal Care. The protocol (HSe-3191) was approved by the Animal Care Committee of the University of Ottawa.

### 4.2. Study Design

#### 4.2.1. Pubertal LPS-Prebiotic Model

We first examined the acute and enduring effect of pubertal LPS exposure and prebiotic intake on the gut microbiota and immune system. In total, 72 mice were used in this experiment. The timing of puberty was determined in mice by examining the first pubertal event or vaginal opening [100]. Mice were categorized into two groups (n = 36): 1—prebiotic group; receiving AHCC (2 g/kg BW/day) in drinking water and 2—control group; receiving regular drinking water for one week before puberty. At puberty (5 weeks of age), half of the mice in each group were injected intraperitoneally (IP) with LPS at a dose of 1.5 mg/kg, which has been shown to provoke inflammation and dysbiosis [6], and the other half were injected with 1X sterile phosphate-buffered saline (PBS) (NaCl 2.7 mM KCl 10 mM Na_2_HPO_4_, 1.8 mM mM KH_2_PO_4_, pH7 (Sigma Aldrich, Oakville, ON, Canada). LPS solution was prepared by dissolving 0.2 mg/mL lyophilized LPS from *Escherichia coli* O26:B6 (Sigma Aldrich, Oakville, ON, Canada) into sterile PBS. Therefore, we had four experimental groups after LPS/PBS injection (n = 18 in each group): 1—control; receiving regular drinking water and injected with PBS, 2—LPS; receiving regular drinking water and injected with LPS, 3—prebiotic; receiving AHCC in drinking water and injected with PBS, and 4—prebiotic + LPS; receiving AHCC in drinking water and injected with LPS. Eight hours after the LPS/PBS injection, half of the mice in each group were sacrificed to study the acute effect of treatment on gut microbiota and immune system. For the remaining mice, nutritional intervention continued for one week after injection, after which the mice received a standard diet until early adulthood. Mice were sacrificed at nine weeks of age and the required samples were collected to study the lasting effects of treatment on the immune system. AHCC^®^ is a standardized extract of cultured *Lentinula edodes* mycelia, produced by Amino Up Co., Ltd. (Sapporo, Japan). 

#### 4.2.2. Pubertal LPS-Probiotic Model

Based on the results from the first experiment, we conducted another experiment to study the durable effect of pubertal LPS exposure and probiotic intake on the gut immune system. Thirty-six mice were categorized into two groups (n = 18): 1—probiotic group; receiving SV-53 (10^9^ CFU/mL) in drinking water, and 2—control group; receiving regular drinking water for one week before puberty. At puberty, half of the mice in each group were injected with LPS or PBS as stated above. Thus, we had four experimental groups after LPS/PBS injection (n = 9 in each group): 1—control; receiving regular drinking water and injected with PBS, 2—LPS; receiving regular drinking water and injected with LPS, 3—probiotic; receiving SV-53 in drinking water and injected with PBS, and 4—probiotic + LPS; receiving SV-53 in drinking water and injected with LPS. One week after injection, the nutritional intervention was terminated and mice were sacrificed at nine weeks of age. The bacterial culture and preparation have been explained elsewhere [56]. Figure 6 illustrates the study design of the pubertal LPS-prebiotic model (acute and enduring experiments) and pubertal LPS-probiotic model (enduring experiment).

### 4.3. Determination of Cytokine Concentrations in the Small Intestine of the Mice by ELISA and Luminex Multiplex Assay

To examine the IL-17A, IL-17F, TGF-β, IL-1β, IL-6, IL-23, and IL-10 levels in the intestinal tissue, after washing the small intestine with 1X PBS, small parts of the ileum (20–25 g) were cut into very small pieces using surgical blades (Fisher Scientific, Toronto, ON, Canada) and collected in microtubes containing lysis buffer (20 mmol/L Tris-HCl (pH 7.5), 150 mmol/L NaCl, 0.05% Tween-20) and a cocktail of protease inhibitors (AEBSF, Hydrochloride, Millipore Sigma, Oakville, ON, Canada). After vortexing and incubation on a rocker at 4 °C for 20 min, homogenized samples were centrifuged at 14,000× *g* for 10 min at 4 °C, and supernatants were collected. The concentrations of total proteins in tissue lysates were measured by the BCA method using Pierce BCA Protein Assay Kit (Thermo Fisher Scientific, Toronto, ON, Canada) and adjusted to 2 mg/mL using the same lysis buffer. Then, the multiplex assay was conducted using Mouse Th17 Panel Magnetic, MTH17MAG-47K (Millipore Sigma, Burlington, VT, USA) and TGFBMAG-64K MAG Bead Kit (Millipore Sigma, Burlington, VT, USA) based on the manufacturer’s instruction, and plates were read using a MAGPIX^®^ System (Millipore Sigma, Burlington, VT, USA). The IL-17A levels were assayed using an ELISA kit (Invitrogen, Vienna, Austria) and the results were calculated by dividing each sample’s observed concentrations of cytokines by the total protein concentration of that sample.

### 4.4. Determination of p-STAT, STAT3, and FOXO1 Levels in the Small Intestine of the Mice by Western Blotting

Small segments of ileum were cut into very small pieces and lysed in the appropriate amount of RIPA buffer (Thermo Fisher Scientific, Toronto, ON, Canada) and the Protease and Phosphatase Inhibitor Cocktail (Thermo Fisher Scientific, Toronto, ON, Canada) for 1 h on a rocker at 4 °C. The protein lysates were collected by centrifuging homogenized samples at 14,000× *g*, for 20 min, at 4 °C, and total protein concentrations were measured using the Pierce BCA Protein Assay Kit (Thermo Fisher Scientific, Toronto, ON, Canada) following the manufacturer’s protocol. The equal amount and concentration of protein samples were run on the 12% Bis-Tris Mini Protein Gels (Invitrogen, Toronto, ON, Canada), using MES SDS Running buffer (Life Technologies, Toronto, ON, Canada) at 200 v for 22 min and then transferred to PVDF membrane in a Trans-Blot Cell (Bio-Rad, Hercules, CA, USA) at 100 v for 1 h. Membranes were incubated with anti-FOXO1 (1:1000), anti-STAT3 (1:1000), anti-phospho-STAT3 (phospho Y705) (1:1000), and anti-β-actin (1:1000) primary antibodies (Abcam, Toronto, ON, Canada) at 4 °C overnight. Then, blots were incubated with horseradish peroxidase-conjugated secondary antibodies (1:10,000) (Jackson Immuno Research Laboratories, West Grove, PA, USA) at room temperature for 1 h. Bands were visualized by chemiluminescence assay using ECL substrate (Bio-Rad, Mississauga, ON, Canada) and quantified by the Bio-Rad Image Lab 6.0.1 Software using β-actin as loading control.

### 4.5. Determination of miRNAs Expression in the Small Intestine of the Mice by Real-Time Quantitative Reverse Transcription PCR (RT-qPCR)

Small pieces of ileum were stored in RNAlater Stabilization Solution (Invitrogen, Carlsbad, CA, USA) for 24 h at 4 °C and then stored at −80 °C until RNA extraction. The total RNA of samples was extracted using miRNeasy mini kit (Qiagen, Toronto, ON, Canada) and their concentrations and purity were determined by a NanoDrop 2000 (Thermo Scientific, Waltham, MA, USA). To assay the expression of selected miRNAs, first, a reverse transcription reaction was undergone to synthesize cDNA using miRCURY LNA RT Kit (Qiagen, Toronto, ON, USA), and then the expression levels of miR-145 and miR-425 were measured by RT-qPCR using hsa-miR-145-5p and hsa-miR-425-5p miRCURY LNA miRNA PCR assay primers (Qiagen, Toronto, ON, Canada) and miRCURY LNA SYBR Green PCR Kit (Qiagen, Toronto, ON, Canada) in a CFX 96 real-time PCR detection system (Bio-Rad, Laboratories, Hercules, CA, USA). Expression of miRNAs was normalized to SNORD65 (mmu) miRCURY LNA miRNA PCR Assay (Qiagen, Toronto, ON, Canada) as the reference gene. The relative expression level of miRNAs was calculated using the ΔΔCT method.

### 4.6. Methylome-Wide Profiling and Data Analysis

In total, 15–20 g of mice ileum samples were homogenized by an electrical homogenizer (Bead Mill 24, Fisher Scientific, Canada) in tubes containing 500 µL cell lysis buffer and 1.5 µL proteinase K. The Gentra Puregene Tissue Kit (33 g) kit (Qiagen, Toronto, ON, Canada) was used to extract tissues’ DNA according to the manufacturer’s instruction. The extracted DNA was then diluted with DNA rehydration solution, provided with the kit, to the final concentration of 20 ng/µL and stored at −20 °C. The concentration of extracted DNA was measured using a Qubit 4 instrument (Thermo Fisher Scientific, Canada). For methylome-wide profiling, 500 ng of extracted DNA was subjected to bisulfite conversion using the EZ DNA Methylation kit (Zymo Research, Irvine, CA, USA). Then, 250 ng bisulfite-modified DNA was analyzed using the Infinium Mouse Methylation BeadChip arrays, which allow for the simultaneous assessment of DNA methylation at more than 285,000 CpG sites (Illumina Inc., San Diego, CA, USA). Methylome-wide data were analyzed using the methylkey pipeline developed by the Epigenomics and Mechanisms Branch at the International Agency for Research on Cancer (https://github.com/IARCbioinfo/methylkey (accessed on 21 January 2023)). Briefly, raw data files were pre-processed, quality control was conducted, and normalization was performed by Noob normalization using the SeSAMe package [101]. Intergroup comparisons were conducted using linear regression analysis as implemented in the limma R package [102]. Regional analysis to identify differentially methylated regions was conducted using the DMRcate package [103].

### 4.7. Gut Microbiome Analysis

For microbiome analysis, the feces samples of mice were collected in sterile microtubes and stored at −80 °C following snap-freezing in liquid nitrogen. Microbiome analysis was conducted by IARC, Lyon, France, using amplicon-based sequencing of V3 and V4 variable regions of the 16S rRNA gene according to the 16S metagenomics sequencing library preparation protocol [104].

### 4.8. Statistical Analysis

Statistical analysis for all experiments except for DNA methylation and microbiome analysis was conducted by GraphPad Prism 5.0 Software (GraphPad Software Inc., San Diego, CA, USA). The distribution of data was assayed by the Shapiro–Wilk test. One-way analysis of variance (ANOVA) followed by Tukey’s post hoc test was performed to compare the means of experimental groups. Data are reported as mean ± SEM of three independent experiments and *p*-value < 0.05 indicates a statistically significant difference between groups. For microbiome analysis, the 16S analysis was conducted by Qiime2, while for differentially abundant taxa comparison, the DESeq2 package for analysis and Wald test were used. Differences in alpha diversities (α-diversity) were assessed by the Chao1, observed species, Shannon and Simpson indexes, and Kruskal–Wallis pairwise test. Sample beta diversity (β-diversity) clustering was assessed using weighted UniFrac PCoA and PERMANOVA pairwise test. Graphs related to α-diversity and taxa abundance were created by GraphPad Prism. Corrected/adjusted *p*-value < 0.05 was considered statistically significant. Differentially methylated genes were defined with false discovery rate (FDR)-adjusted *p*-value < 0.05 and absolute inter-group beta value difference of >0.05. For pathway visualization, KEGG pathway enrichment analysis was performed using Enrichr. Figure 6A,B was generated by BioRender.com (accessed on 19 September 2023).

## Figures and Tables

**Figure 1 ijms-24-14610-f001:**
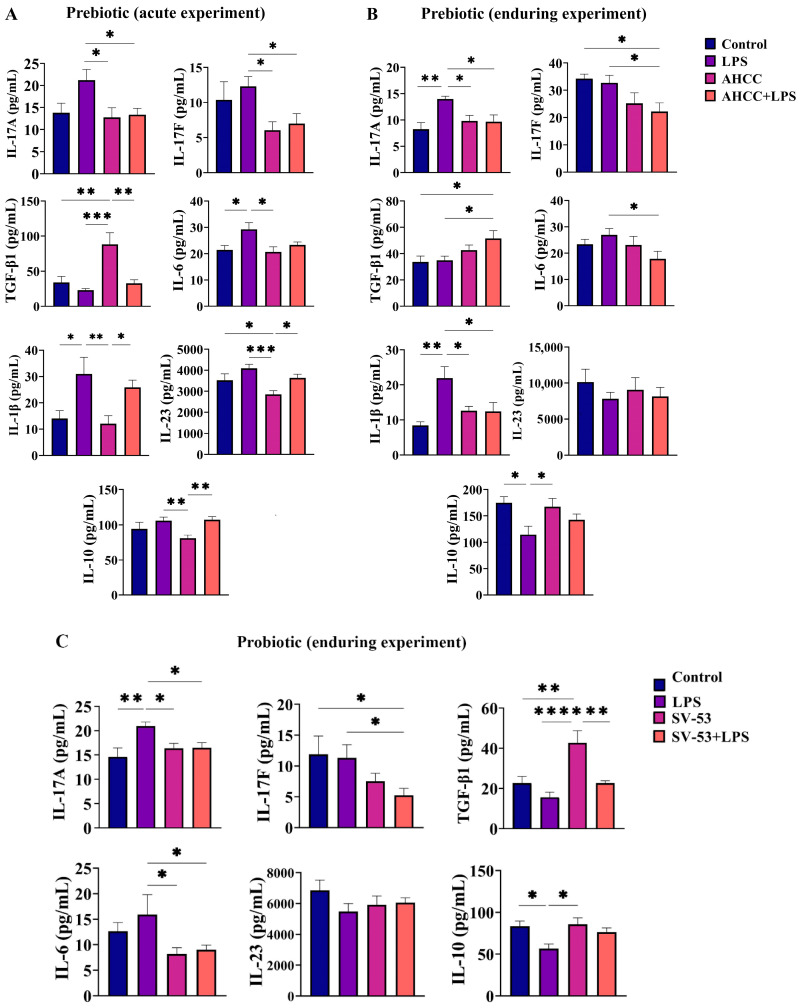
(**A**) concentrations of IL-17A, IL-17F, TGF-β, IL-6, IL-1β, IL-23, and IL-10 in the small intestine of mice 8 h after injection of a single dose of LPS at puberty. Mice received AHCC (2 g/kg BW/d) in drinking water or drinking water without AHCC for one week before injection. (**B**) concentrations of IL-17A, IL-17F, TGF-β, IL-6, IL-1β, IL-23, and IL-10 in the small intestine of adult mice four weeks after injection of a single dose of LPS at puberty. Mice received AHCC (2 g/kg BW/d) in drinking water or drinking water without AHCC for two weeks, one week before, and one week after LPS injection. (**C**) concentrations of IL-17A, IL-17F, TGF-β, IL-6, IL-23, and IL-10 in the small intestine of adult mice four weeks after injection of a single dose of LPS at puberty. Mice received SV-53 (10^9^ CFU/mL) in drinking water or drinking water without SV-53 for two weeks, one week before, and one week after LPS injection. One-way ANOVA and Tukey’s post hoc tests were used to compare groups. All values are mean ± SEM. * *p* < 0.05, ** *p* < 0.01, *** *p* < 0.001, and **** *p* < 0.0001.

**Figure 2 ijms-24-14610-f002:**
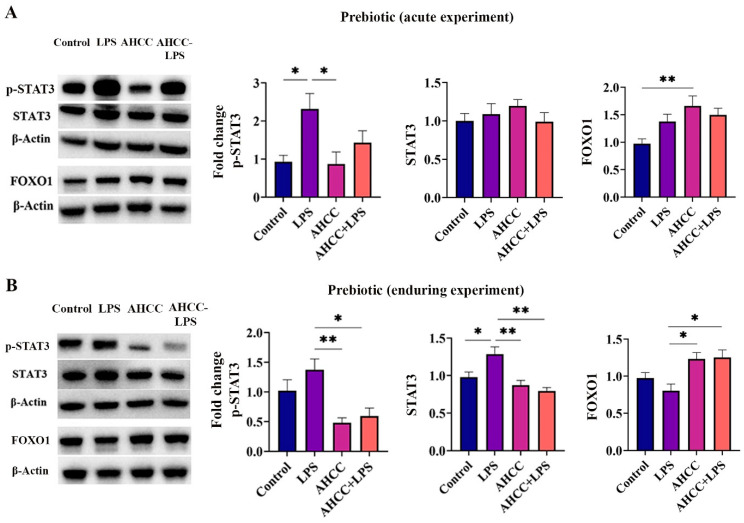
(**A**) levels of p-STAT3, STAT3, and FOXO1 in the small intestine of mice 8 h after injection of a single dose of LPS at puberty. Mice received AHCC (2 g/kg BW/d) in drinking water or drinking water without AHCC for one week before injection. (**B**) levels of p-STAT3, STAT3, and FOXO1 in the small intestine of adult mice four weeks after LPS injection at puberty. Mice received AHCC (2 g/kg BW/d) in drinking water or drinking water without AHCC for two weeks, one week before, and one week after LPS injection. (**C**) levels of p-STAT3, STAT3, and FOXO1 in the small intestine of adult mice four weeks after LPS injection at puberty. Mice received SV-53 (10^9^ CFU/mL) in drinking water or drinking water without SV-53 for two weeks, one week before, and one week after the LPS injection. One-way ANOVA and Tukey’s post hoc tests were used to compare groups. All values are mean ± SEM. * *p* < 0.05, and ** *p* < 0.01.

**Figure 3 ijms-24-14610-f003:**
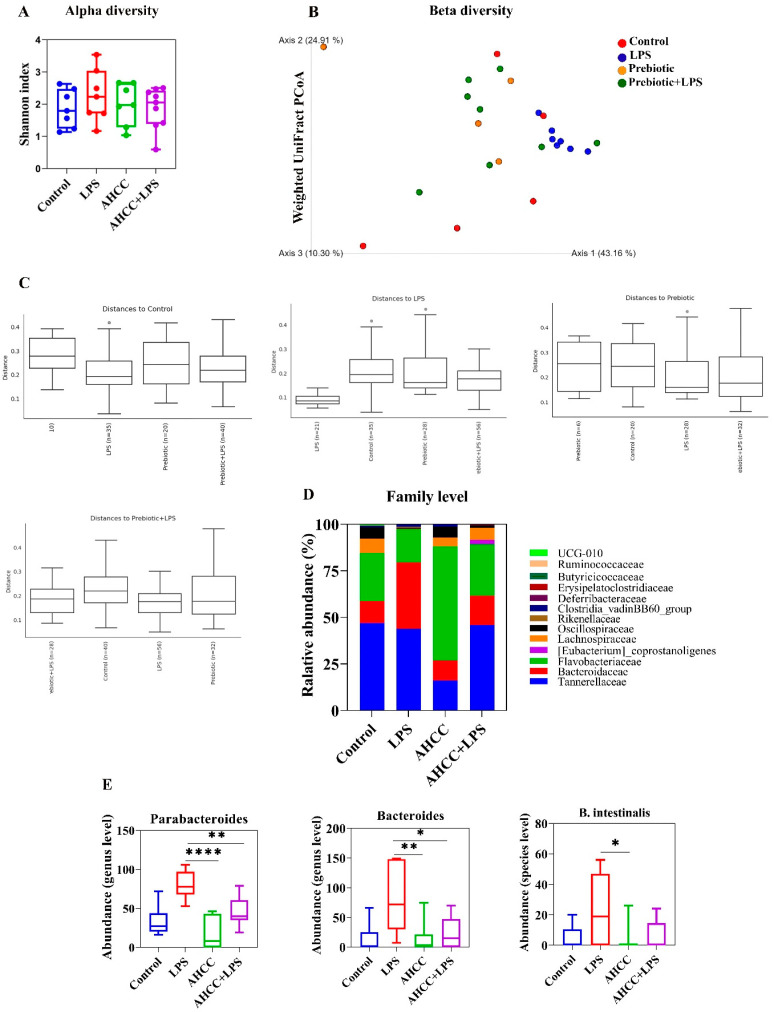
Gut Microbiome analysis of pubertal mice. Four-week-old female Balb/c mice received AHCC (2 gr/kg BW/d) in drinking water or drinking water without AHCC for one week before LPS injection. At 5 weeks of age, mice were injected with LPS and 8 h after injection, mice were sacrificed, and feces samples were collected. (**A**) alpha diversity, (**B**) weighted UniFrac PCoA, (**C**) weighted UniFrac distance, (**D**) relative abundance of most abundant taxa at the family level, and (**E**) differentially abundant taxa at the genus and species levels. All values are mean ± SEM. * *p* < 0.05, ** *p* < 0.01, and **** *p* < 0.0001.

**Figure 4 ijms-24-14610-f004:**
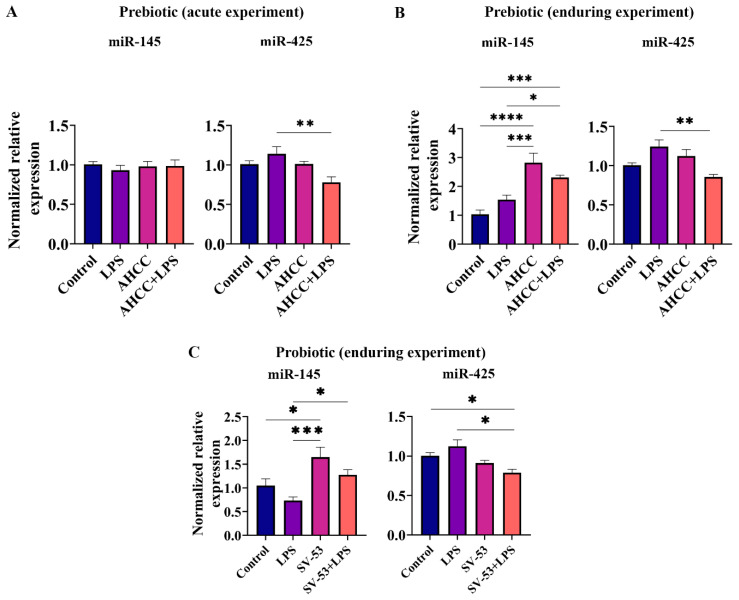
(**A**) relative expressions of miR-145 and miR-425 in the small intestine of mice 8 h after injection of a single dose of LPS at puberty. Mice received AHCC (2 g/kg BW/d) in drinking water or drinking water without AHCC for one week before injection. (**B**) relative expressions of miR-145 and miR-425 in the small intestine of adult mice four weeks after LPS injection at puberty. Mice received AHCC (2 g/kg BW/d) in drinking water or drinking water without AHCC for two weeks, one week before, and one week after LPS injection. (**C**) relative expressions of miR-145 and miR-425 in the small intestine of adult mice four weeks after LPS injection at puberty. Mice received SV-53 (10^9^ CFU/mL) in drinking water or drinking water without SV-53 for two weeks, one week before, and one week after the LPS injection. One-way ANOVA and Tukey’s post hoc tests were used to compare groups. All values are mean ± SEM. * *p* < 0.05, ** *p* < 0.01, *** *p* < 0.001, and **** *p* < 0.0001.

**Figure 5 ijms-24-14610-f005:**
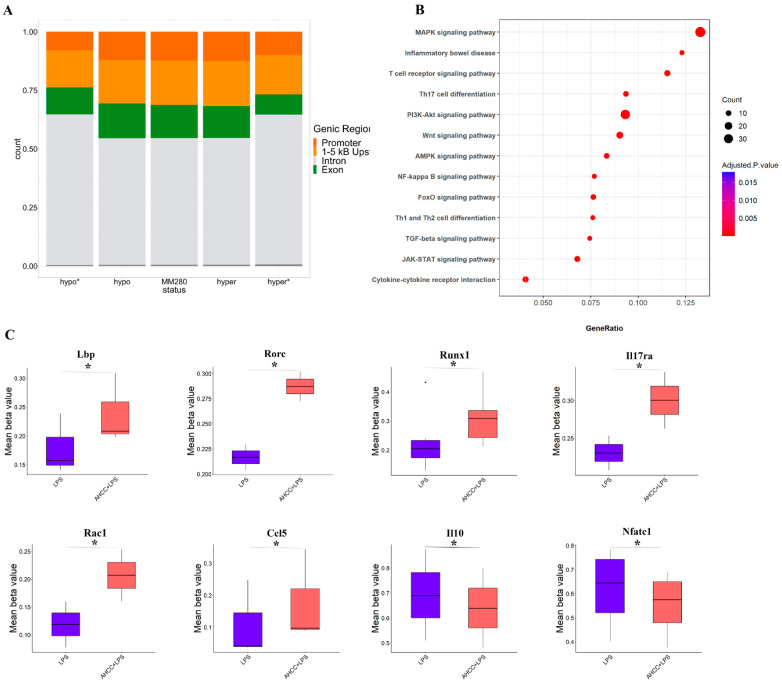
DNA methylation analysis in adult mice four weeks after LPS injection at puberty. Mice received AHCC (2 gr/kg BW/d) in drinking water or drinking water without AHCC for two weeks, one week before, and one week after LPS injection. (**A**) genomic regulatory regions, where the hypo and hyper columns indicate all hypo- and hypermethylated regions, respectively, and hypo* and hyper* columns represent significant hypomethylated and hypermethylated DMRs, respectively. MM280 column represents the overall distribution of the array. (**B**) pathways enrichment analysis visualized by Enrichr, (**C**) boxplots of differentially methylated genes. * *p* < 0.05 vs. LPS.

**Figure 6 ijms-24-14610-f006:**
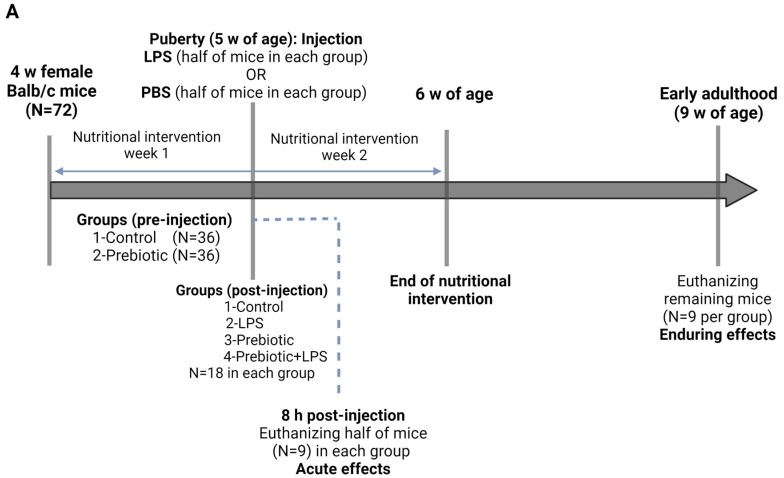
Study design showing the experimental timeline, groups, LPS injection, and dietary intervention for (**A**) the pubertal LPS-prebiotic model (acute and enduring experiments) and (**B**) the pubertal LPS-probiotic model (enduring experiment).

## Data Availability

The data presented in this study are available on request from the corresponding author.

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
