# Peer review of "Lentinula edodes Cultured Extract and Rouxiella badensis subsp. acadiensis (Canan SV-53) Intake Alleviates Immune Deregulation and Inflammation by Modulating Signaling Pathways and Epigenetic Mechanisms"

_ijms, 2023, doi:10.3390/ijms241914610_

Round 1
Reviewer 1 Report
It is a complex, difficult to understand in some points, experimental work.
I would like the authors to explain why they decided to treat mice with a prebiotic or a probiotic not being in common use; and why they do not study the combined treatment of pre- and probiotic. However, in conclusion, line 654, they say "while prebiotic and probiotic intake may mitigate these negative consequences "; "AND" means together, it is better to use "OR".
introduction, paragraph 2 may be shortened, since it includes, more or less common knowledge.
From line 101 and thereafter, up to the end of introduction authors are referred to the aim of the study. On lines 101-104 they "aimed to investigate whether exposure to LPS in the pubertal window results in immune dysfunctioning ... and whether dietary intervention ... can block LPS-induced immune deregulation later in life.". On lines 111-116 they say "Our main objective was to investigate whether prebiotic and probiotic intake could mitigate LPS-induced inflammatory responses by regulating ...cytokines, .. and miRNAs ...function at the gut level ." What is exactly the aim of the study? I understand it is dual, but must be written more comprehensively, in few words - one sentence - and not in two paragraphs.
Section 2.2.1 . A table presenting the groups would be helpful to understand instant of a long description. The same for section 2.2.2.
Line 548, ref [13] is underlined
Author Response
Dear Editor and Reviewer,
Thank you so much for processing and reviewing our manuscript.
The order of the manuscript’s section was modified based on IJMS requirements.
In addition, reviewer 1 comments have been responded to in the attached file and revised and highlighted in the manuscript.
Please find attached file for the author's response.
Thank you so much for your time and consideration.

Reviewer 2 Report
The manuscript (MS) has focused on the pertinent subject matter, gathering substantial data and presenting findings on the impact of prebiotics and probiotics on immunomodulation. The results obtained are indeed satisfactory and hold significant relevance.
- The keywords have too many abbreviations; it is better to replace or rewrite them.
- Use a similar style to represent one type, e.g. IL-17 (line 31) and IL17 (line 35), and address the issue in the entire manuscript.
- Regarding line 42, remove the term "may."
- In line 136, provide details about the strength of the PBS used.
- For improved comprehension, consider creating a schematic diagram to accompany the methodology section.
- In an effort to streamline the manuscript, it is recommended to reduce the number of references, which currently stands at 104. Avoid the practice of citing multiple sources to support a single statement, as exemplified by [1,28,29] (line 87); [6,36,58] (line 472); [38,60-62] (line 482). Consolidate references where possible.
Moderate english editing
Author Response
Dear Editor and Reviewer,
Thank you so much for processing and reviewing our manuscript.
The order of the manuscript’s section was modified based on IJMS requirements.
In addition, the reviewer’s comments have been responded to in the attached file and have been revised and highlighted in the manuscript.
Please find attached file for the author's responses.
Thank you so much for your time and consideration.

Round 2
Reviewer 2 Report
The asked changes has been added in the manuscript. I recommend for the publication of the article.
NA